Arbuthnott and Hajat *Environmental Health* 2017, **16**(Suppl 1):119

# The health effects of hotter summers and heat waves in the population of the United Kingdom: a review of the evidence

Katherine G. Arbuthnott[1,2*] and Shakoor Hajat[1]

## Abstract

It is widely acknowledged that the climate is warming globally and within the UK. In this paper, studies which assess the direct impact of current increased temperatures and heat-waves on health and those which project future health impacts of heat under different climate change scenarios in the UK are reviewed.

This review finds that all UK studies demonstrate an increase in heat-related mortality occurring at temperatures above threshold values, with respiratory deaths being more sensitive to heat than deaths from cardiovascular disease (although the burden from cardiovascular deaths is greater in absolute terms). The relationship between heat and other health outcomes such as hospital admissions, myocardial infarctions and birth outcomes is less consistent. We highlight the main populations who are vulnerable to heat. Within the UK, these are older populations, those with certain co-morbidities and those living in Greater London, the South East and Eastern regions.

In all assessments of heat-related impacts using different climate change scenarios, deaths are expected to increase due to hotter temperatures, with some studies demonstrating that an increase in the elderly population will also amplify burdens. However, key gaps in knowledge are found in relation to how urbanisation and population adaptation to heat will affect health impacts, and in relation to current and future strategies for effective, sustainable and equitable adaptation to heat. These and other key gaps in knowledge, both in terms of research needs and knowledge required to make sound public- health policy, are discussed.

**Keywords:** Climate change, Heat, Summer, Heat-wave, Temperature, Mortality, Morbidity, United Kingdom, Adaptation

## Background

There is a well-established link between increased ambient temperatures and adverse health outcomes. This has been demonstrated for the United Kingdom (UK) and across a range of settings [1, 2]. It is also unequivocal that the global climate is warming and the consensus is that this is largely due to anthropogenic causes [3]. Recent probabilistic projections [4] using a medium emissions scenario estimated that summer mean temperatures in some locations in southern England are very likely to rise between 2.2 and 6.8 °C (10% and 90% probability levels respectively), with a central estimate of 4.2 °C (50% probability level) by 2080 relative to a 1961–1990 baseline [4]. The frequency of extreme heat events or heat waves is also projected to increase [5, 6]. Therefore, understanding both the current and future impact of hot weather on health and the measures which can be taken to reduce these impacts is important for planning and carrying out effective public health action.

The aim of this narrative review is to bring together evidence from epidemiological studies and health impact assessments to provide an overview of what is known about current and projected effects of heat on population level health in the UK. We review only those impacts directly related to increased ambient heat exposure (i.e. evidence of the indirect impacts of temperature changes such as changing patterns of infectious diseases are not included). This review focuses on UK studies, though evidence from other high-income settings has occasionally been included for illustration and discussion where evidence from the UK is lacking.

* Correspondence: Katherine.arbuthnott@lshtm.ac.uk
[1]Faculty of Public Health and Policy, London School of Hygiene and Tropical Medicine, 15-17 Tavistock Place, London WC1H 9SH, UK
[2]Chemicals and Environmental Effects Department, Centre for Radiation, Chemical and Environmental Hazards, Public Health England, Chilton, Oxon OX11 0RQ, UK

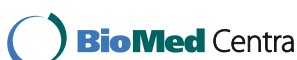

Lastly we discuss what is known about the potential for adaptive measures to reduce the impact of heat on health and identify key gaps in knowledge.

## Methods

Studies were included which used data from the United Kingdom, aggregated to regional, conurbation (e.g. Greater London) or national level. These could have been as part of a larger study including data from other countries. We included studies in the epidemiological and impact assessment part of the review which estimate:

- current or past associations between a) patterns of hot weather (time series or case-crossover studies) and health outcomes (see below) and b) heat waves (studies using episode analysis) and health outcomes
- impacts of warming under climate change (past and projected) on health.

All outcomes related to health and well-being were considered (e.g. mortality, indicators of morbidity – use of health services e.g. hospital admissions, NHS direct calls, GP visits, specific outcomes such as myocardial infarction etc.). Where outcomes for specific sub-groups of the population (e.g. older persons) were available, these were included to further understand potential vulnerable sub-sets of the UK population.

In order to locate the relevant epidemiological studies we searched the database Ovid Medline using terms relating to two main concepts: one relating to climate and high temperatures/heat waves/climate and one relating to health outcomes. The search terms were combined using appropriate Boolean operators and results limited to the English language and humans. No date restrictions were applied and searches were updated in January 2017. Searches were not initially restricted by geographical region, as many studies included data from multiple countries. However, from these multi-country studies we specifically review results from the UK. In order to capture relevant articles not indexed in the databases, we also snowballed references.

## Epidemiological studies of the effect of heat on health

### The effects of ambient heat on mortality and morbidity
#### Study design

In general, the relationship between mortality (and morbidity) and changes in ambient temperature has been assessed using time series regression models or case crossover designs, which have been shown to yield similar results [7, 8]. UK based studies have generally shown a U,V or J shaped relationship between temperature and health outcomes, with an increased risk of mortality or morbidity above and below a given thresholds or minimum mortality temperatures (MMTs). Results are presented as the increased risk of outcome (relative risk (RR) for time series studies, odds ratios (OR) for case-crossover studies) for every 1 °C increase above that threshold.

Studies presented here have controlled for a combination of potential time varying confounders, such as pollution – typically ozone or $PM_{10}$ (though the precise role of pollutants in analysis is currently under debate [9]), public holidays and day of the week, in addition to season and long term trends. Most studies have used a lag of between 0 and 2 days to analyse the relationship between exposure and outcome as, in contrast to the effect of cold, the effect of heat on health generally occurs within a few days of exposure [10].

The specific aspects of each study, including details of methods, exposure and outcomes and threshold selection are summarised in Additional file 1: Table S1 (UK based studies examining the effects of increased temperatures on mortality and Additional file 2: Table S2 (UK based studies examining the effects of increased temperatures on morbidity).

### Effect of ambient heat on mortality

Across the UK there is an increased risk of all-cause mortality with increasing temperature above threshold values [10–30](see Additional file 1: Table S1 for details), a finding consistent with studies from a variety of global settings [1, 2]. The size of the effect varies between regions and also between studies, which may be partly driven by modelling choices such as the choice of the threshold but also due to local population differences in climate, demography and social factors. Modelled above the 93rd percentile of maximum daily temperatures, one recent study found the increase in all-cause mortality is 2.1% for each 1 °C increase in temperature (95% CI:1.6%, 2.6%) for England and Wales [22]. Some UK based studies have examined the attributable deaths due to heat exposure. For example, Gasparrini et al. [22] found that 1.0% of all summer deaths in England and Wales were attributable to temperature and Armstrong et al. [21] found a similar figure.

Most UK studies examining cause-specific mortality [13, 20, 22], have categorised deaths as cardiovascular, respiratory or external (to retain power). Within these categories, respiratory and external causes of mortality have been demonstrated to be most sensitive to heat in terms of RR (though due to the total larger number of deaths occurring from cardiovascular disease, a greater portion of heat attributable deaths are due to cardiovascular causes). However, a more detailed breakdown of causes of death [22] (see Additional file 1: Table S1 for further details) found that within these general categories (e.g.

cardiovascular, respiratory deaths) some causes of death were much more sensitive to heat (in terms of magnitude of RR) than others (for example the RR of death from atrial fibrillation or pulmonary heart disease was much greater than for cardiovascular disease in general) and also that endocrine, nervous system and genitourinary causes of death are sensitive to heat . The risk of suicide is also increased by heat [17].

Baccini et al. analysed Years of Life Lost (YLL) due to heat in a multi-city European study [24]. An average of 1914 heat-related YLL per year in London was estimated, not adjusted for harvesting or mortality displacement (see below).

### Mortality displacement

Mortality displacement ("harvesting") refers to the process by which some deaths which occur in an already frail population are brought forward by a few days or weeks. The effect of mortality displacement on the estimated risk of mortality caused by increased ambient temperatures has been specifically examined for London.

For example, the 1914 heat-related YLL in London, estimated by Baccini et al. [24], reduced by 81% to 356 per year when adjusted for short-term harvesting. One study specifically examined the effects of mortality displacement across a range of settings [14] and found that for London, the RR of all cause and cardiovascular mortality did not persist past 2 days and that by day 11 the sum of the effects for all-cause mortality was 0 (the risk for respiratory cause mortality, however, remained over the 28 day period tested). In contrast, the excess risk of heat on mortality persisted for 3 weeks in Delhi. This was attributed to heat causing death in children or people who are not usually at risk of imminent death. More recently, annual time series were used to examine longer time mortality displacement for London. However, the study was unable to draw any firm conclusions about how long heat deaths may have been brought forward by, due to the imprecision of estimates [31].

Of note, the harvesting evaluation could be sensitive to modelling choices. For example, most studies use a threshold fitted from data from the general population, whereas the threshold may be lower for those at risk of harvesting.

### Effect of ambient heat on morbidity and other indicators of health and wellbeing

In the UK, there is no clear evidence of an association between total emergency hospital admissions and raised ambient temperatures [32]. However a significant increase for respiratory admissions (5.4% (95% CI: 1.9%, 9.1%)) increase per 1 °C above the 23 °C threshold (and for renal disease admissions (1.3% increase per 1 °C above the 18 °C threshold (95% CI: 0.3%, 2.4%)) has been

demonstrated [32]. These findings have been supported by other work [33]. The contrast between mortality and hospital admissions may in part be due to heat causing a rapid deterioration in those who are vulnerable, meaning they do not present to medical attention before death. There is evidence from two studies of an association between paediatric emergency trauma admissions and increasing ambient temperature in England [34, 35]. For example, Atherton et al. report an 11% increase (95% CI: 12%, 38%) in paediatric trauma admissions for every 5 °C increase in maximum temperature (and larger increases using minimum temperatures as an exposure variable) [34]. This association for paediatric admissions was greater than for adult trauma admissions, which was found to be non-significant in the study by Atherton et al. and of smaller magnitude the study by Parsons et al. (1.8% increase in adult trauma admissions for 5 °C increase in temperature, significance not reported) [35].

At a daily resolution, there has been no demonstrated increased risk of myocardial infarction (MI) admissions with increasing temperatures in the UK [36]. However, an increased risk in MI (1.9% (95% CI: 0.5%, 3.3%) per °C above the threshold) has been demonstrated 1–6 h after heat exposure [37]. This excess-risk at shortest lags was followed by reduced risk at 24 h leading to the hypothesis that the reduction at longer time intervals is due to short term displacement.

A recent study in Birmingham [38] demonstrated correlation between increased temperatures and poorer ambulance category A response times. Further work is needed to quantify the risk, for example using time series regression, to include better confounder (e.g. number of ambulances on duty in a day, air pollution etc.) and seasonal control in the analysis.

In London, although the odds of preterm birth were affected by seasonality (increased odds in winter months), no association between premature birth and increased or decreased temperatures up to six days before birth was found [39]. Studies examining different exposure timings and neonatal outcomes of interest (e.g. birth weight, pre-term birth) within the UK would be welcome as results from other settings have yielded mixed outcomes [40].

The effects of heat on these indicators of health and wellbeing are summarised in Table S2 in the supplementary materials.

### Heat waves

The effect of heat waves or defined periods of high temperatures on health has been mostly examined using episode analyses, where the outcomes over an identified heat-wave period are compared to an expected baseline. There is no consensus around the definition of a heat

wave used within the UK, though most studies include a duration and severity component. For example, the World Meteorological Organization define a heatwave as "when the daily maximum temperature of more than five consecutive days exceeds the average maximum temperature by 5°C, the normal period being 1961-1990" [41]. However, criteria used to trigger public health warnings in Public Health England's heat wave plan, use region-specific thresholds based on maximum daytime and minimum night-time temperatures to trigger different alert levels [42].

Given the lack of a consistent definition of a heatwave (leading to the use of different duration and severity components used in analysis), differences in baseline selection for episode analyses, differences of heatwave timing within the summer season (heatwaves and hot days occurring earlier in the summer may have a greater effect on mortality [43, 44]) and differing characteristics of the preceding winter (there is evidence that winter mortality may affect heat related mortality of the following summer [45]), the effects of heatwaves between years within the UK is difficult to compare.

### Mortality and heat wave episodes

Estimates of excess mortality have been made for many of the notable heatwave periods in the UK in recent years [11, 46–49].

The most severe single heat wave (HW) of recent times was in 2003, for which the health effects were greatest in continental Europe [50]. In England between 4-13th August 2003, the Central English Temperature (CET) exceeded average values (from 1871 to 2000) by 8 °C. This HW period in 2003 HW has been calculated to have led to 2091 excess deaths in England - a 17% increase (95% CI: 15%,19%) above the expected baseline [48].

Recently Green et al. used consistent definitions (any period that would trigger a Met Office heatwave alert of level 1, or a single day when mean Central England Temperature was greater than 20 °C) to compare total daily excess deaths occurring during all the defined heatwave periods over each summer between 2003 and 2013 [49]. The estimated excess mortality of all heatwave periods occurring in 2013 was considerably lower than in preceding years. For example, in 2013 there were an estimated total 195 excess deaths (95% CI: -87,477) across all heatwave days in those older than 65 years, with an excess of 10 deaths per heatwave day (95% CI: -4, 24). By comparison, for the same age group (65 plus years) the number of excess deaths for all heatwave days in 2003 was 2234 (95% CI: 1936, 2532) with 102 excess deaths per heatwave day (95% CI: 88,115) and in 2006 was 2323 (95% CI: 2008,2638) for all heatwave days with an average of 89 excess deaths (95% CI: 77,101) per heatwave day. There are many possible explanations for

the decrease in later years. It is possible that the population has become less vulnerable to the effect of HWs (for example the Heatwave Plan was introduced in 2004 in England). In addition, the lower peak temperatures occurring in 2013 compared to 2003 and 2006, the timing of heatwaves in the season (see above) and differences in other environmental exposures such as pollution and humidity may have contributed.

In separate studies, notable increases in mortality have also been demonstrated for the 1995 and 1976 heatwaves [11, 47]. The 1995 heatwave is estimated to have caused an increase in mortality of 8.9% (95% CI: 6.4%, 11.3%) over England and Wales (and a 16% increase in Greater London) and the 1995 heatwave is estimated to have led to a 30.7% (95% CI: 25.8%, 35.8%) increase in mortality in Greater London.

### Mortality displacement

Whilst none of the studies described here undertook a formal analysis of harvesting for UK data, estimates attributed less than 10% and 25% of the excess deaths to harvesting in the Paris 2003 [51] and Chicago 1995 [52] heat-waves respectively. Of note, there is current debate about how best to calculate mortality displacement for heatwave periods [53].

### Heat waves and other health outcomes

In contrast to the large number of excess deaths in the 2003 heatwave, the proportional increase in hospital admissions over the whole of England was small – just 1% (95% CI: 1%,2%). This is consistent with findings that ambient heat increases mortality risk, but not all cause hospital admissions. London suffered a higher increase in admissions compared to the rest of the country in the 2003 heatwave (with a 6% proportional increase across the age groups, though this too was smaller than the 42% increase in mortality seen in London during the same heatwave period) [48]. A similar pattern has been demonstrated for the 1995 heatwave, where an excess mortality of 10.8% (95% CI: 2.8%, 19.3%) was calculated, but the increase in hospital admissions of 2.6% (95% CI -2.2%, 7.6%) was not significant [32].

Recently the use of syndromic surveillance demonstrated an increase in a range of selected indicators during heatwave periods in July 2013 [54]. These indicators included calls to NHS direct and GP services and visits to the emergency department for heat/sun stroke symptoms. A moderate increase in NHS direct calls was seen in the 2003 heatwaves, but was only significant for the younger age groups (0–4 years and 4–14 years) [55]. Other indicators for GP consultations and emergency department attendance for asthma, severe wheeze, myocardial infarction and cerebrovascular accidents showed no increase during the 2013 heatwave [56]. Indeed, the

number of GP consultations over this period decreased for myocardial infarctions.

During the 1976 HW in Birmingham [57] there was no significant increase in sickness benefit claims but a modest increase was seen in GP consultations.

### Work productivity and heat

Although the impact of hot weather has not been assessed in terms of productivity or occupational health outcomes in the UK, there is evidence that these outcomes are affected by heat elsewhere. A review of occupational health impacts of heat exposure, including data from a range of countries, found that manual workers exposed to extreme heat or working in hot environments and especially those in tropical low-middle income countries are at increased risk [58]. At risk occupations included farming, construction work, firefighters, manufacturing (where workers are exposed to heat generation) and those in the military. A recent review of all the heat related occupational fatalities in the US between 2000 and 2010 found a yearly death rate of 0.22 per million workers. Those most at risk included men, those working in agriculture and construction and Hispanics [59].

### Distribution of health impacts by geographical region, age, gender and socio-economic status

#### Geographical/regional differences

A consistent finding, is that the increase in risk of heat related mortality is greatest for London, the South East and the East of England. Typically London, the South East and East of England have the highest RR per °C rise in temperature (typically thresholds for these studies have been set at a given percentile of the regional temperature distribution, so that all effects are estimated above say the 93rd percentile but the absolute value of this varies between locations, with hotter regions having higher absolute values) [20–22, 26, 60]. Bennet et al. [25] examined geographical vulnerability at greater resolution (district level) for England and found the same general pattern of risks of heat related mortality being higher in the South of England but also identified districts at particular high risk, for example the London borough of Tower Hamlets.

London, the South East and East of England also have the highest proportion of excess mortality calculated during heat waves. During the 2003 heat wave, the largest increase in proportion of deaths was seen in London (42%), followed by Eastern England (27%) compared to lowest proportional increases of 4% in the North West and 2% in the North East [48].

Reasons for the difference in risk between regions are not known, but differences in demographics, social and economic factors may play a part, as may the hotter

temperatures experienced – particularly during heatwaves. It is likely that for cities such as London, the Urban Heat Island (UHI) has a role. The UHI refers to the increased temperature within built-up urban areas compared to surrounding suburban rural areas [61, 62], due to alterations in the energy balance as a result of surface properties (e.g. albedo etc.), land use and city design. Estimates from the West Midlands attribute around 50% of the deaths in the 2003 heatwave to the UHI effect [63] and it is likely to become increasingly important as we see an increasing proportion of the population living in cities [64].

### Age

In the UK, older age groups are more at risk of heat-related morbidity and mortality [19, 25, 65]. This may be due to diminished ability to thermo- regulate, increasing medical co-morbidities or use of medications, and social factors which may limit behavioural adjustments to the increased temperatures. Vulnerability in older age groups is increasingly important, as demographic projections show an ageing UK population [66]. The exception to this is within populations with chronic psychiatric conditions (including psychoses, dementia and alcohol and substance misuse). In this subgroup of the population, those under the age of 65 were at increased risk [23].

### Sex

In UK based studies females (in particular older females) appear to have a higher risk of dying at hotter temperatures and in a heat wave [16, 19, 57, 67]. However, it is possible that many of these results for females are confounded by age as this finding is not consistent across countries [68, 69].

### Underlying co-morbidities and medications

In the UK, patients with neurological and psychiatric diagnoses, such as dementia and substance misuse and those prescribed antipsychotics, antidepressants and hypnotics have been shown to be at increased risk of heat related mortality [23]. An increased risk of hospitalisation and mortality from use of prescribed medication and recreational substance use has been found elsewhere [70, 71]. Plausible mechanisms underlying this include decreased thirst and decreased sweating [72]. No UK studies have examined heat related risk of mortality (or morbidity) in those with cardiovascular, respiratory, renal disease and diabetes, though evidence from other settings suggests they may be more vulnerable to the effect of heat [73, 74].

### Socio-economic status

UK studies have not found a consistent relationship between socio-economic status (SES) (quintiles of deprivation analysed) and an increased vulnerability to heat or heat waves, analysed with data aggregated to regional or district level [16, 25]. However, analysis even at district level, could mask potential differences in vulnerability between socio-economic groups occurring at the individual level.

Studies in other countries, most notably North America have found some linkage with SES [73, 75]. This relationship may be partially explained or related to access to air conditioning, which is much more prevalent in the US compared to the UK or by differences in access to healthcare which are less prevalent in the UK (due to universal coverage provided by the National Health System).

### Other factors affecting vulnerability

In the UK, place of residence (nursing or residential home) has been associated with an increased risk of heat-related mortality [16, 46], though this may simply reflect that those populations are likely to be more frail. There is little evidence of the effect of heat being modified by living alone or in a flat [16].

## Current heat related impacts attributable to climate/weather and observed climate change

Despite observed warming of the climate over recent years [4], there is no evidence of a substantial increase in heat related mortality in the UK. Indeed, studies would suggest that there has either been a decline in heat related mortality in the UK over the last century [15] and between 1971 and 1997 in South East England [12], or that heat related risk has remained unchanged in London between the two time periods of 1996–2002 and 2004-2010 [28] and between 1993 and 2006 [30]. One study demonstrated a marginal increase in heat related mortality between 1977 and 2005 (of 0.7 deaths per million population per year [76], see below). Potential explanations for the decrease or lack of change in heat related mortality, despite an ageing population and observed warming, include a contribution from both specific adaptive policies (such as the introduction of the heatwave plan in 2004 [42]) and spontaneous changes not specifically aimed at adapting to heat but which may reduce vulnerability to it, such as improved healthcare and improved standards of living (this may be particuarly relevant to the decrease in heat related mortality observed over the early part of the last century [15]).

Christidis et al. [76], using optimal detection, investigated the contribution of climate (anthropogenic and natural influences) and adaptation (reduced population vulnerability to heat, rather than specific adaptive measures) to changes in mortality occurring between 1976 and 2005 in England and Wales. Results indicated a small increase in heat related mortality over the time period (of 0.7 deaths per million population per year) but they estimated this increase would have been higher (1.6 deaths per million population per year) if no adaptation had taken place. Results of the optimal detection analysis also demonstrated that if no adaptation had taken place, then anthropogenic influences on climate would have been the main influence on heat related mortality.

Recently, Mitchell et al. demonstrated that anthropogenic climate change increased the risk of excess heat-related mortality in the 2003 heatwave by around 20% in London [77].

## Projected future health impacts of heat

Health impact studies estimating future heat-related mortality in the UK [26, 60, 78–82] have projected increases in heat related mortality throughout the twenty-first century. Health impact studies for a wide range of countries have been reviewed elsewhere [83].

These health impact studies typically take an epidemiological relationship, as described in the studies above, and apply future temperature projections (often derived from regional climate projections), to this relationship. To our knowledge, no current published estimates use temperature projections based on the more recent Representative Concentration Pathways [84]. Uncertainty in these health impact studies is introduced at many levels including uncertainty in the baseline co-efficient (see above sections), future emissions, the parameters, processes and initial conditions included in climate change models, and a populations adaptive capacity [79]. Regarding the uncertainty introduced by climate projections, mortality estimates for London were found to be most sensitive to climate model physics uncertainty compared to emissions scenario choice and downscaling uncertainty [80]. Some [85] have suggested that future health impact projections are more sensitive to the choice of climate projections rather than uncertainties in the baseline exposure-response relationship or demographic projections.

The most recent projections of future impacts of climate change on UK mortality, estimate an increase in heat related mortality from a baseline of 1974 deaths per year in the 2000s, to 3281 deaths per year (66% increase) in the 2020s, 7040 deaths per year (257% increase) in the 2050s and 12,538 deaths (535% increase) in the 2080s. These projected deaths are attributable both to increased temperature and the increased projected population size. This study [60], based partially on the UK Health Protection Agency (HPA) climate change risk assessment [78], used a baseline co-efficient from the

time series analysis of historic weather and mortality data in England and Wales from 1993 to 2006. The climate data was based on the UKCP09 climate projections for a medium emissions scenario. These medium emissions scenario UKCP09 projections used the Met Office Hadley Centre Regional Climate Model (HadRM3-PPE-UK) to dynamically downscale Global Climate Model (GCM) results for historical emissions and the Special Report Emissions Scenario (SRES) A1B. Though demographic changes were considered for the risk assessment, assumptions about adaptation were not. The regions projected to be most at risk are the East and South of England, the West Midlands and London. The additional heat-wave effect was significant only for London, where it was estimated to add an extra 58%, 64%, 70% and 78% to heat-related mortality in the 2000s, 2020s, 2050s and 2080s [60].

Of studies that have projected heat-related mortality for European or global cities which include London [26, 79, 81, 82], one included estimates which accounted for population adaptation [79]. The GCM used was the UK HadGM3 (with no downscaling) and emissions scenarios were A2 and B2. Adaptation assumptions included an assumption that the population would not adapt, adapt to a 2 °C or to a 4 °C increase in mean temperatures (achieved by shifting the dose response curve so that the heat-slope remained unchanged but the threshold temperature is increased). The projected mortality for London roughly halved for each 2 °C adaptation assumed.

In another study the proportion of respiratory hospital admissions attributable to heat for Northern Europe was projected to increase from 0.13% (lowest estimate 0.10%, highest estimate 0.15%) to 0.27% (lowest estimate 0.19%, highest estimate 0.32%) in the period 2021–2050 [86]. One less recent Department of Health report [87] projected in-patient hospital days associated with increased temperatures to increase to 285,000 per year by 2050 under a medium emissions scenario (using the UKCIP98 climate projections) compared to the baseline of 81,000 per year in 1995 and 1996 for England and Wales. Up to date projections using more recent data, climate projections and taking advantages in improved statistical techniques for modelling the underlying baseline associations between hospital admissions and heat, however, are lacking.

## Potential for impacts of heat to be avoided by adaptation measures

Adaptation has often been used to refer to planned and unplanned structural and policy level actions which may reduce a population's vulnerability to heat. The extent to which adaptation can be achieved will depend upon the

local context, vulnerabilities and adaptive capacity [88]. Acclimatisation more commonly refers to increased short term physiological tolerance to heat. It is outside the scope of this review to present evidence on physiological acclimatisation.

There is evidence that populations residing in areas with a warmer climate have increased thresholds for heat sensitivity [18, 89, 90] and that vulnerability to heat has decreased over time in several locations [12, 30, 91, 92]. However, identifying specific adaptive measures which have contributed this, and distinguishing the role of these from demographic or socio-economic background factors which may decrease sensitivity to heat, is challenging (though some studies have indicated a correlation between decreasing vulnerability and increased prevalence of air conditioning [91]). Further, the evidence of efficacy for given planned and implemented policies or interventions to specifically reduce negative health outcomes such as mortality and morbidity, using robust methods, is scarce [93, 94].

Potential adaptive strategies range from interventions and actions at an individual and housing level, health systems and infrastructure level through to national policy and plans. However there are complex interactions between impacts of many given policies. It is important to consider whether any of policies could be 'maladaptive' or have unintended consequences (e.g. [95, 96]). It would be most desirable if strategies reduced health effects of both heat and cold (for example through improved building design, which may also be useful for climate change mitigation [97–99]) and had other health co-benefits (e.g. urban greening – see below).

Adaptive measures with the potential to reduce the health impacts of heat are briefly considered here from the individual level to National Policy level.

### Behavioural measures

Behavioural measures to protect against the effect of heat such as use of cool clothing, increasing intake of non-alcoholic fluids, and restricting strenuous activity to cooler parts of the day, are often advised [42, 100]. However there is a lack of studies that have fully quantitatively evaluated the impact of behavioural measures on heat related outcomes.

The efficacy of behavioural measures will depend both upon the effectiveness of the intervention to reduce heat exposure and also the willingness of individuals to take up these behaviours: one recent survey examined a number of protective behaviours carried out during the 2013 heatwave and home characteristics (e.g. prevalence of air conditioning) in a sample of the UK population [101]. It found that the elderly were less likely to partake in some personal and home protective measures but were more likely to open their windows at night. Higher

income earners were also more likely to engage in personal and home protective measures. It therefore highlighted groups for further targeting of health messaging.

### Interventions at an individual or place of residence level
#### Air conditioning and electric fans
Prevalence of air conditioning in the UK is low (estimated at 3% in a recent survey [101]) and evidence for its efficacy in reducing heat-health impacts is not available within the UK. However, there is evidence, mostly from Northern America, that air conditioning offers protection against the impacts of heat on health [74, 102] and that seeking a public space with air conditioning can be protective [103]. Indeed this is a measure included in many heat wave plans. However, there are potential disadvantages to promoting this as an adaptive strategy. For example there may be equity issues in terms of both the prevalence and availability of air conditioning units. Access to shared air-conditioned space may remove some of these concerns. Extensive use of air conditioning could increase energy demands and itself contribute to the urban heat island effect [96]. There are also examples of power grid outages (which can occur during extreme weather events such as heat waves), which would limit the protection that air conditioning could offer [104]. Further evidence is required in order to be able to effectively weigh the benefits of air conditioning against its environmental, energy and health costs. Passive cooling measures may reduce the need for air conditioning.

A systematic review [105] of the evidence on use of electric fans found no studies which met its inclusion criteria (study type of randomised controlled trials or other experimental designs, such as interrupted time series studies). It therefore concluded that there were not enough high quality studies to assess the evidence. Where case control studies were found, results for the protective effect of fans were inconclusive with some studies showing a protective effect and others the opposite.

#### Community, services and Urban Design level adaptations
Many interventions have been postulated that may improve adaptation at an urban design level, for example increased use of green spaces [106], improved insulation of housing against heat and cold [98], improved surface properties of buildings (e.g. increasing albedo) [107] etc. Some of these, such as increased green spaces can provide co-benefits to health [108, 109] in addition to adapting to climate change [106]. Unintended consequences of interventions should also be considered (e.g. indoor air quality concerns in more air tight houses etc. [110], exposure to allergens and volatile organic compounds released from urban trees [111]). Further, health

services will also need to adapt to ensure comfortable and reliable care in warmer temperatures (e.g. indoor temperatures, power supply in heat waves, ability to cope with potential increased demands during hot weather [112]). These measures will require multi-sectorial working and planning.

### Policy level interventions
#### Heat warning systems (heatwave plans)
Public Health England publishes an annual heatwave plan [42], which has been operational since 2004, following the 2003 European heatwave. The plan details actions that individuals and organisations can take to reduce the risk of heat to public health. These include both high level preparations and actions for health and social care organisations in England and advice for the general public on health protection and specific measures to protect vulnerable groups. As such there are several core parts of the plan ranging from strategic planning and preparedness to the alert system and advice on communication with the public. It details cascades of heatwave alerts and specific advice on protective measures (for example key messages for the public include keeping out of the heat, taking measures to cool down and keeping the environment cool and monitoring others who are vulnerable). The heat health alert service detailed in the plan splits heatwave alert levels into 5 categories: level 0 (long term planning which is to take place all year round) to level 4 (a major incident – severe or prolonged heatwave which affects several National level sectors in addition to health). Temperatures used to trigger heatwave alerts are set at regional specific thresholds and use daytime and night time maximum temperatures. For example the daytime maximum temperature which would trigger an alert in London is 32 °C compared to 28 °C for the North East of England.

A recent systematic review has examined the effectiveness of Heat Warning Systems (HWS) [113] (any action plan based on meteorological and demographic information typically including early alerts and public health protection measures, tailored to local contexts). The review found that research in this area is limited. Six out of the 15 studies included found some evidence of effectiveness of HWS, but were unable to robustly establish a causal relationship between HWS implementation and reduced mortality and morbidity. Further evidence to evaluate both the individual recommendations within the plans and the effectiveness of the plans overall would be useful for improvements to these.

### National level frameworks for adaptation
Under the Climate Change Act of 2008 [114], a framework was established for the UK to respond to climate

change. As part of this, government is required to assess the risks from climate change to the UK and prepare strategies to address these (e.g. through the National Adaptation Plan) and to increase resilience to climate change.

### Targeting policy

Attention should also focus on ensuring the adaptation needs of the most vulnerable populations are met. However, this may be complex to achieve.

Identifying those at risk has the potential to improve targeted prevention policies. Some UK studies have sought to incorporate known vulnerabilities into spatial heat health risk assessments combining geographical information on the heat hazard, exposure and vulnerability of small area populations within cities. Examples include the development of a vulnerability index for London [115], a spatial heat health risk assessment for Birmingham [116] and a conurbation-scale heat risk assessment undertaken for Greater Manchester [117], combining projected temperature rises with urban vulnerability indicators. These vulnerability mapping studies are often undertaken to support decision and policy making. However, one recent systematic review, demonstrated that although this is often the authors stated intention, there is currently little demonstrable evidence of their use to influence policy [118]. This highlights the importance of ensuring that research is both useable and relevant to policy makers.

Identification of vulnerable populations can also be difficult to translate into the social and medical support required. Firstly, whilst front-line responders feel they know those individuals who are at increased risk at a given time, vulnerabilities such as co-morbidities and social isolation fluctuate rapidly and keeping a systematic record of these is difficult [119]. Further, barriers to implementation of protective policies may result from the low priority given to heat risk by practitioners and also by the vulnerable populations themselves [119]. The picture may in fact be made more complex by reinforcement of social norms within at risk populations. For example increased contact in social networks may perpetuate views amongst the elderly that they are not at risk [120].

### Conclusions

This review has discussed the key findings of UK studies illustrating the association between increased temperatures and heat waves and mortality. Results for other health outcomes are less consistent. Vulnerable populations include the elderly and those with certain co-morbidities. Future projections all predict an increase in heat related deaths throughout this century, the magnitude of which depends on assumptions used within models.

However, key gaps in our knowledge remain and warrant further investigation. These include knowledge gaps around the relationship between health outcomes and more extreme temperatures not yet experienced in the UK, a better understanding of certain vulnerabilities to heat (for example, the effect of UHIs and different individual and local level factors which increase or decrease risk within the UK) and the effect of high temperatures on other well-being outcomes (such as economic productivity).

Considering future impacts, the use of temperature projections based on the newer Representative Concentration Pathways (RCPs) [84] could be examined for future health impact assessments. Further work on impact by certain groups such as by age and socio-economic status in addition to location would be useful. There are also many gaps in our knowledge about adaptation. For example, what works now and what may work in the future? How can past specific adaptive measures be identified and evaluated? And how might adaptive strategies translate from one setting to another? How can we best model adaptation in future heat-health impact assessments? There are gaps in our knowledge around adaptive measures that might have multiple health benefits and how public health can best work with other sectors to promote integrated research and policy development in this area. The design and use of suitable indicators for adaptation and ensuring equity components are reflected in these, is important in order to evaluate progress.

Addressing these knowledge gaps will be paramount to ensure that evidence can inform policy for appropriate actions and use of resources to minimise future health impacts of heat, and to improve health under a changing climate.

### Additional files

Additional file 1: Table S1. UK based studies examining the effects of increased temperatures on mortality. (DOCX 27 kb)

Additional file 2: Table S2. UK based studies examining the effects of increased temperatures on morbidity. (DOCX 19 kb)

Additional file 3: Open peer review. (PDF 50 kb)

### Abbreviations
CET: Central England Temperature; CI: Confidence Interval; DOH: Department of Health; GCM: Global Climate Model; GDP: Gross Domestic Product; IPCC: Inter-governmental Panel on Climate Change; MI: Myocardial Infarction; MMT: Minimum Mortality Temperature; ONS: Office for National Statistics; PM: Particulate matter; RA: Risk assessment; RCP: Representative Concentration Pathways. These represent four greenhouse gas trajectories and describe used in the fifth IPCC report (AR5). Rather than describing emissions scenarios (e.g. like the SRES), the product is a set of four pathways that lead to a range of radiative forcing levels from 2.6–8.5 W/m2. These endpoints are compatible with the range of previous emissions and climate policy scenarios; SRES: Special Report Emissions Scenarios. Produced by the IPCC, these are baseline scenarios and include the A1 family (an integrated world with rapid economic growth) A1F1 (emphasis on fossil fuels) A1B1 (balanced emphasis on all energy sources) A1T (emphasis on non -fossil energy sources). A2 (more heterogeneous world, self-reliant nations, increasing

populations, regionally orientated economic development). B1 (rapid economic growth but towards a service and IT economy, population rises then declines past 2050, introduction of clean and efficient technologies) B2 (world more divided/heterogeneous but more ecologically aware); UHI: Urban Heat Island; UK: United Kingdom; UKCP09: UK Climate ProjectionsUKCP09 represent the 5th generation of climate projections for the UK. These provide probabilistic projections for three different future emissions scenarios, representing low, medium and high emissions (chosen from scenarios in the Special Report on Emissions Scenarios (SRES)); UKCP98: UK Climate ProjectionsUKCP98 represent the 3rd generation of climate projections for the UK. These used the HadCM2 model. They presented four scenarios 'low' (low forcing, low climate sensitivity), Medium-low, medium-high and high (high forcing, high climate sensitivity); YLL: Years of Life Lost

### Acknowledgements
This review was funded under the "Living With Environmental Change (LWEC)" programme. KA is supported by the Public Health England PhD studentship scheme. The authors would like to thank two anonymous reviewers for their comments, which have improved this review article substantially and Sotiris Vardoulakis and Clare Heaviside for their very helpful comments on a draft of the manuscript.
This paper is a reduced version of a technical paper provided in support of a Health Report Card produced for the UK Living With Environmental Change (LWEC) Network.

### Funding
Publication of this article was funded by the UK Living With Environmental Change (LWEC) Network. LWEC was succeeded in 2016 by the Research and Innovation for our Dynamic Environment (RIDE) Forum (http://www.nerc.ac.uk/research/partnerships/ride/).

### Availability of data and materials
Not applicable

### Open peer review
Peer review reports for this article are available in Additional file 3.

### About this supplement
This article has been published as part of Environmental Health Volume 16 Supplement 1, 2017: Special Issue on the impact of climate change on health in the UK. The full contents of the supplement are available online at https://ehjournal.biomedcentral.com/articles/supplements/volume-16-supplement-1.

### Authors' contributions
KA reviewed the literature and drafted the manuscript, SH provided critical input to the manuscript. Both authors read and approved the final manuscript.

### Ethics approval and consent to participate
Not applicable

### Consent for publication
Not applicable

### Competing interests
The authors declare that they have no competing interests.

##

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
