## [Open peer review. (PDF 50 kb) · Environmental Health]

Reviewer reports

Title: The health effects of hotter summers and heat waves in the population of the United Kingdom: a review of the evidence

Reviewer 1: Daniel Oudin Åström

The current paper is a review of the evidence between high ambient temperatures / heat waves and different health outcomes in the UK. The topic of this review is timely and of public health interest. The results may be useful when discussing the impact of heat today, impacts of future climate change on human health and adaptive capacity of populations. However, there are limitations to the study.

Major comments:

My main concern is that no inclusion criteria are presented in the current review of the UK based studies. The review would benefit from letting the reader know how the literature search was conducted. For instance, what terms were used to identify relevant studies, when the studies should have been published and what kind of studies that were included.

In addition, the literature search is incomplete. I am missing a number of publications where the impact of heat on mortality in the UK were investigated, albeit in a multicity setting. For instance, de Donato et al (2014), Gou et al (2014) and Gasparrini et al (2015). The current review sometimes includes multicity studies, but not in a consistent way. Presenting all such studies would improve the quality of the review considerably.

It would be easier to get an overview of the studies included in the review were included in a table where the key information such as study setting and thresholds were presented.

Minor comments:

R 54-55: Mortality and morbidity from many causes is too unspecific, please specify.

R 75: I believe the referred study used daily maximum temperature with a lag of 0-1 days.

R 92: What was the threshold here?

R 93: Other UK-based studies ... only refers to one study though.

R 116: References missing.

R 167: Again, one study.

R 178 and 180: Increased risk of mortality or hospital admissions?

R 219. Which studies?

R 265: Why was reference 75 not included in the section on ambient heat and mortality?

Other:

R 105: ... between increased TEMPERATURE and poorer ambulance ...

The section on work productivity and heat could be complemented with the review by Gubernot et al (2015) in American journal of industrial medicine.

Socio-economic status: I think this section could be improved by a discussion regarding individual level SES versus neighbourhood level SES. See also Urban et al in IJERPH (2016) for another study in European setting.

R 216: There is a nice study from the Netherlands regarding evidence of declining vulnerability to heat covering more than a century (Ekamper et al 2009, Demographic Research). This would provide additional information in a European setting as compared to the cited US studies. In addition, Gasparrini recently published a study on the declining effects of temperature on mortality within summers.

MMT is not found in the main document.

Competing interests declaration: I declare that I have no competing interests.

Reviewer 2: Simon Gosling

I have added line numbers to the manuscript (attached) and refer to them here throughout the review.

I like the idea of this review, which aims to summarise the impact of current increased patterns of high temperatures and heat waves on health as well as studies that have projected future health impacts of heat under different climate change scenarios, for the UK. The research could, potentially be a very useful resource.

However, despite my enthusiasm for a piece of work such as this, it is my opinion that the review requires major revision before it can be published and of use to researchers (and a policy audience in particular). The overarching reason for this recommendation is that I find the review to be significantly deficient in its depth of and breadth of literature coverage. Numerous articles are missing from the review and to this end the article does not achieve its aim. Sadly this means that the article does not provide an accurate summary of the current state of science on UK heat and mortality studies (present and projected). In order to address this, the authors need to conduct their review far more systematically and in turn refer to, and discuss, the literature that is missing.

I have provided some general and specific comments below which I hope that the authors will consider and find constructive. Addressing these comments will unfortunately require a significant amount of reading and reviewing, and new writing, to be undertaken by the authors, but I believe that if these comments are addressed then the overall quality of the manuscript will be improved greatly.

General comments

The title does not mention anything about health. The immediate impression to the reader is that this is going to be a review of climatological evidence. To enhance the chances of this article, if it is published, showing up in academic search engine results it might be worth changing the title to better reflect that contents of the review.

Numerous articles that I expected to see in the review are missing, so I am curious to know the inclusion/exclusion criteria used for selecting articles. There should really be a short Methods section that describes how the review was conducted. I acknowledge that the authors note in the Introduction that they only include epidemiological studies, but the problem with doing this is that it misses out a very large amount of evidence that is crucial, and needed, to actually address the authors' aim. This is one of the reasons that the review does not accurately represent the current state of knowledge on heat and health in the UK.

As a result of very little information on how the review was conducted, the reader is left asking several important questions, which really need to be answered so that the reader can understand the context and quality of the review:

- What tools were used for conducting the literature review (e.g. PubMed, Google Scholar, Web of Science)?

- What key search terms were used?
- Were any journals included/excluded?
- Did the authors only search for articles that focus on the whole of the UK, or did they include studies that investigated impacts for smaller spatial domains, e.g. regions, counties, cities? And were global-scale studies included where data for the UK could be extracted?
- What years of publication did the authors limit their searches to?

A selection of the many articles that I expected to see cited which are missing are listed below. This is certainly not a comprehensive list of missing articles (I don't have time to list all of them unfortunately – a comprehensive and systematic literature search would yield them). I suggest that the authors consider including the articles listed below (and others) and moreover that they revise their review so that it is a more complete representation of the current state of science on this topic. Conducting the review with a more systematic and rigorous approach is really needed to address this. In its present form, the manuscript reads more like an informal ad-hoc review that may have been included as part of a report. I doubt that this was the authors' intention.

- Christidis et al. (2010) Causes for the recent changes in cold- and heat-related mortality in England and Wales. *Climatic Change* 102: 539-553.
- Gosling et al. (2012) The benefits of quantifying climate model uncertainty in climate change impacts assessment: an example with heat-related mortality change estimates. *Climatic Change* 112: 217-231.
- Gosling et al. (2007) Climate change and heat-related mortality in six cities Part 1: model construction and validation. *International Journal of Biometeorology* 51: 525-540.
- Green et al. (2016) Mortality during the 2013 heatwave in England – How did it compare to previous heatwaves? A retrospective observational study. *Environmental Research* 147: 343–349.
- Hajat et al. (2005) Mortality displacement of heat-related deaths: a comparison of Delhi, São Paulo, and London. *Epidemiology* 16:613–620.
- Heaviside et al. (2016) Attribution of mortality to the urban heat island during heatwaves in the West Midlands, UK. *Environmental Health* 15.
- Lindley et al. (2006) Adaptation Strategies for Climate Change in the Urban Environment: Assessing Climate Change Related Risk in UK Urban Areas. *Journal of Risk Research* 9: 543-568.
- McGregor et al. (2007) The social impacts of climate change. Environment Agency. ISBN 9781844328116.
- Smith et al. (2016) The impact of heatwaves on community morbidity and healthcare usage: a retrospective observational study using real-time syndromic surveillance *Int. J. Environ. Res. Public Health*, 13: 132.
- Smith et al. (2015) Estimating the burden of heat illness in England during the 2013 summer heatwave using syndromic surveillance. *J Epidemiol Community Health*. doi:10.1136/jech-2015-206079.
- Watkiss et al. (2009) Impacts of climate change in human health in Europe. PESETA-Human health study. EC Joint Research Centre.
- Wolf et al. (2015) On the Science-Policy Bridge: Do Spatial Heat Vulnerability Assessment Studies Influence Policy? *Int. J. Environ. Res. Public Health* 12: 13321-13349.

Unfortunately, what is more concerning, as a result of the lack of literature coverage, is that there is a risk of some statements and conclusions in the article being factually inaccurate because the authors have overlooked key studies. A couple of examples of this occurring are:

1. Lines 213-214 where the authors state that “The lack of UK studies which have estimated the burden of health effects of heat considered attributable to climate change that has already occurred represents an area for further research”. This is not the case because attribution studies on heat and health *have* been done for England and Wales (Christidis et al. 2010) but sadly the authors seem unaware of this area of research.
2. The situation described around lines 323-330 with regards to policy level interventions is far more complex and nuanced than the authors suggest. A highly relevant discussion of this is presented by Wolf et al. (2015; *Int. J. Environ. Res. Public Health* 12: 13321-13349) but no reference to this is included (see also Lindley et al. 2006; *Journal of Risk Research*).

Harvesting (i.e. mortality displacement) is mentioned in several different parts of the manuscript. Some readers might find it helpful if a specific section was included on harvesting that draws all of this information together.

Throughout the manuscript the text weaves between referring to UK and non-UK studies. This is somewhat unhelpful as the abstract suggests the aim is to focus on the UK. This is highlighted in lines 42-48, which is also a little confusing - are the authors saying here that they tried not to include studies that were: 1) written by authors who are based at non-UK institutions (e.g. a study on heat and health in London but written by authors from the Netherlands); or 2) written about health impacts in non-UK countries (e.g. a study about heat and health in the Netherlands)? If they mean the former, then why would they try to extrapolate to the UK? This would not be appropriate. And why review such studies anyway, if the aim is to focus on the UK? Lines 186-188, for example, also drift away from the UK. Overall, the jumping between UK and non-UK studies in the manuscript results in a feeling of inconsistency in approach to the review. I suggest that the authors re-evaluate their overall approach used to include/exclude studies.

Specific comments

Line 33: Need a brief description of why there is a range in projected temperature change (it is quite a large range). Also double-check whether the warming quoted really is relative to the last century – the values appear to be relative to pre-industrial (1800s), not last century (1900s).

Lines 22-23: This sentence is somewhat ambiguous. Need to better differentiate here whether the increases in mortality are due to population change only, climate change only, or climate change and population change with the majority of the change being attributable to population change.

Line 35: General comment: It already is acknowledged as being very important, e.g. see the health chapter in the forthcoming UK Climate Change Risk Assessment.

Lines 62-63: It would be helpful to indicate what some of the other confounders tend to be, beyond just air pollution.

Lines 64-66: Considering the content of this section of text it seems odd that the authors do not mention the term Relative Risk (RR).

Line 65: It is worth noting that threshold temperatures are also known as “optimum temperatures” and “minimum mortality temperatures”.

Lines 69-70: Fair point here about the contentious issue of defining heatwaves but considering the aim of this paper, does the UK have a formal definition of a heat wave, perhaps referred to in the Met Office heat health warning system, that can be included here? It would be nice to provide a more UK-centric focus here.

Line 75: Please provide some examples of thresholds. Do they vary across the UK or are they generally consistent?

Line 78: most sensitive to heat compared to what other causes of death – respiratory, IHD, CVD, or what?

Lines 79-80: This does not fit with the previous sentence and it effectively contradicts it.

Lines 106-107: Please provide more detail on the critique of this study as it is currently vague – what is the issue with the way that the study incorporated confounding factors?

Lines 121-122: How did mortality observed in these two studies compare to what was observed during the 2003 heatwave?

Line 128: Model of what?

Line 129: This subsection is very short and its value as a standalone section is questionable. I suggest the authors consider merging this with one of the previous sections.

Lines 136-149: A number of global-scale assessments of the impact of climate change on labour productivity have been conducted recently (e.g. Dunne et al. (2013; *Nature Climate Change*) and see also a body of research by Tord Kjellstrom), so it seems a pity not to extract results from these global studies and report them here, especially when there is so little work done on the UK specifically.

Line 144: Why is a US study being discussed when the aim is to review work focussed on the UK?

Line 153: please provide more details here, at the very least some references to support this statement.

Lines 213-214: Worth noting that detection and attribution studies have provided attribution estimates of the contribution of anthropogenic forcings to the 2003 European heatwave (climatologically, not health, however; Stott et al. 2004; *Nature*).

Line 222: this sentence suggests that these studies have ignored other variables, like humidity, which are used in indices such as the apparent temperature. Can the authors please confirm whether all UK studies exclusively use projected temperature only?

Line 225: Suggest changing from “parameters and processes included in climate change models” to “parameters, processes and initial conditions included in climate change models”.

Lines 247: Worth also noting as a caveat that the emissions scenarios used are somewhat “old” now as they used SRES scenarios. It would be nice to see some studies using the more recent RCPs and SSPs for the UK – it’s good to see that this is mentioned in the Conclusion.

Line 248: The authors need to clarify what they mean by adapting to 2C here because it is not clear. Do they mean adapting to an increase in mean summer temperatures of 2C?

Line 253: does the “medium emissions scenario” have a name, e.g. A1B? If it does, it would be nice to provide it, since the previous examples cited in this section have the emissions scenarios named.

Line 277: Could here refer to Part L of the UK Building Regulations, which describes building code standards for the provision of insulation in new dwellings. The U-values (which are used to define the insulation capacity) have essentially improved over several decades, partly in response to climate change projections and the 2050 80% emissions cut target (see the 2008 Climate Change Act). Better

insulation, as defined in Part L, helps to keep houses warmer in winter and cooler in summer. Note, however, that the regulations only apply to new building constructions (see Part L documentation).

Lines 346-347: Another key area on adaptation that needs more research is knowledge/methods on how adaptation can be incorporated and modelled in climate change projection studies of health.

Line 355: The abbreviations table provides lots of details on SRES, describing what each scenario means, but the four RCPs are not explained. Please provide an equal level of detail for the RCPs (details are in the *Climatic Change* Special Issue on the RCPs).

Various lines: some numbers are written without commas, e.g. 200 000 whilst others are, e.g. 200,000. Please be consistent.

Competing interests declaration: I declare that I have no competing interests.

Reviewer reports – 2nd round

Reviewer 1: Daniel Oudin Åström

The revised version of the review of the evidence between high ambient temperatures / heat waves and different health outcomes in the UK is much improved. The authors have expanded the review considerably and also responded to the issues raised by the reviewers.

Minor comments:

My only remaining main comment is regarding the discussion on temperature and morbidity. Recently, Otte im Kampe (2016) published a review on the impact of high temperatures and unintentional injuries where two UK studies, investigating trauma admissions, were included. This section could be expanded on.

R 88: Please reference the statement regarding longer lags for cold.

R 112: One study.

R 113 and R 121: It is presumably the same study being referred to, the YLL paper by Baccini et al, however, two different studies are being referred to. Also, the YLL paper used quite crude measures of YLL.

R 206-208: This sentence needs to be rewritten.

R 220: To be addressed in the gaps of knowledge? It would be interesting to read such a study in a UK setting.

R 232: This stand-alone sentence does not make much sense.

R 285: Why was ref 73 not included in the mortality and heat wave section above?

R286: Living alone

Other comments:

When several studies are being referred to the references are not always in sequential order.

Please be consistent in the use of the number of decimals.

Author response to 2nd round review

Dear Reviewer,

Thank you for reviewing the revised manuscript on the effects of hotter summers and heat waves in the population of the United Kingdom, and for your additional comments. These have been very helpful and we hope that these have now been addressed .

Please find our response (in bold) to your specific comments below. Please note the line numbers highlighted in the responses refer to the 'clean' revised document.

Yours sincerely,

Katherine Arbuthnott

Response to comments

Minor comments:

My only remaining main comment is regarding the discussion on temperature and morbidity. Recently, Otte im Kampe (2016) published a review on the impact of high temperatures and unintentional injuries where two UK studies, investigating trauma admissions, were included. This section could be expanded on. **Thank you for highlighting this. The UK studies have now been included (lines 144-151 and in table S2).**

R 88: Please reference the statement regarding longer lags for cold. **This has been included (line 90-91).**

R 112: One study. **This has been amended (line 110)**

R 113 and R 121: It is presumably the same study being referred to, the YLL paper by Baccini et al, however, two different studies are being referred to. Also, the YLL paper used quite crude measures of YLL. **Thank you, this has been amended (it was indeed one paper) in lines 116 and 124.**

R 206-208: This sentence needs to be rewritten. **The sentence has been rewritten (lines 216 - 219)**

R 220: To be addressed in the gaps of knowledge? It would be interesting to read such a study in a UK setting. **This has been included as a gap in knowledge now – line 493**

R 232: This stand-alone sentence does not make much sense. **This has now been omitted**

R 285: Why was ref 73 not included in the mortality and heat wave section above? **The reference has now been included in the mortality and heat wave section (line 184).**

R286: Living alone. **This has been corrected (line 296)**

Other comments:

When several studies are being referred to the references are not always in sequential order. **Thank you, the ordering of group references are now sequential throughout the text.**

Please be consistent in the use of the number of decimals. **This has been addressed – all numbers in text are now to one decimal place.**